# Structural and Dynamic-Based Characterization of the Recognition Patterns of E7 and TRP-2 Epitopes by MHC Class I Receptors through Computational Approaches

**DOI:** 10.3390/ijms25031384

**Published:** 2024-01-23

**Authors:** Nicole Balasco, Maria Tagliamonte, Luigi Buonaguro, Luigi Vitagliano, Antonella Paladino

**Affiliations:** 1Institute of Molecular Biology and Pathology IBPM-CNR c/o Department Chemistry, Sapienza University of Rome, Piazzale Aldo Moro 5, 00185 Rome, Italy; nicole.balasco@cnr.it; 2Immunological Models Lab, Istituto Nazionale Tumori—Istituto di Ricovero e Cura a Carattere Scientifico (IRCCS)—“Fond. G. Pascale”, Via Mariano Semmola 53, 80131 Napoli, Italy; m.tagliamonte@istitutotumori.na.it (M.T.); l.buonaguro@istitutotumori.na.it (L.B.); 3Institute of Biostructures and Bioimaging IBB-CNR, Via Pietro Castellino 111, 80131 Napoli, Italy; luigi.vitagliano@cnr.it

**Keywords:** MHC, tumor antigens, peptidic epitopes, E7, TRP-2, AlphaFold, MD simulations, molecular recognition

## Abstract

A detailed comprehension of MHC-epitope recognition is essential for the design and development of new antigens that could be effectively used in immunotherapy. Yet, the high variability of the peptide together with the large abundance of MHC variants binding makes the process highly specific and large-scale characterizations extremely challenging by standard experimental techniques. Taking advantage of the striking predictive accuracy of AlphaFold, we report a structural and dynamic-based strategy to gain insights into the molecular basis that drives the recognition and interaction of MHC class I in the immune response triggered by pathogens and/or tumor-derived peptides. Here, we investigated at the atomic level the recognition of E7 and TRP-2 epitopes to their known receptors, thus offering a structural explanation for the different binding preferences of the studied receptors for specific residues in certain positions of the antigen sequences. Moreover, our analysis provides clues on the determinants that dictate the affinity of the same epitope with different receptors. Collectively, the data here presented indicate the reliability of the approach that can be straightforwardly extended to a large number of related systems.

## 1. Introduction

All biological processes rely on stringent intermolecular recognition. These partnerships take place by exploiting a variety of different mechanisms. Indeed, in some cases, the interacting biomolecules are structurally pre-organized for the association while in others flexible ligands bind to a rigid receptor. Particularly intriguing is the mechanism that underlies the exposure of antigens in the immune response, as in this case the binding of a flexible ligand is essential for the stabilization of the receptor. In the cytotoxic immune response, the T-cell receptor (TCR) recognizes antigenic peptides that are bound and presented by the major histocompatibility complex (MHC). Several crystallographic structures of TCR-peptide/MHC complexes show that the TCRs simultaneously interact with both the peptide and the MHC protein.

Major histocompatibility complex class I consists of helical domains as well as a C-terminal membrane anchor region. Before they appear on the outer side of the cell membrane, MHC class I molecules associate with the β2-microglobulin and an antigenic peptide consisting of 8–14 amino acid residues. These peptides are bound in a groove formed by two α-helices (α1 and α2) and a slightly curved β-sheet. Despite the large polymorphic variance among MHC molecules, the fundamental traits of peptide binding are well conserved. The two helices shape a narrow binding cleft that accommodates peptides stabilized by a set of conserved hydrogen bonds between the side chains of the MHC molecule and of the peptide, leading to the occupation of defined pockets. The molecular flexibility of the interacting partners represents a key point in the recognition mechanism, and the stability of MHC class I molecules appears highly dependent on the interaction with the bound peptide [1,2,3,4,5,6]. Indeed, this recognition process relies on the progressive stabilization of the interactors involved. In its general scheme, the binding of the peptide stabilizes the MHC heavy chain (α) receptor by rigidifying its F pocket through the binding of its C-tail. This event is required for the consequential recruitment of the β2-microglobulin before the formation of the entire TCR-peptide/MHC complex in T-cell activation.

A full understanding of the structural features of the MHC-epitope complex formation is of fundamental importance for the design and development of new antigens that could be effectively used in immunotherapy. A major limitation to the comprehension of the structural basis of the MHC-epitope recognition process is the difficulty of making valid generalizations. Indeed, the high variability of the peptide that can be loaded and the huge number of MHC variants present in the organisms make the binding process highly specific. In this scenario, although hundreds of structures of MHC-peptide have been solved and deposited in the Protein Data Bank (PDB), our current knowledge of the process is still limited. Considering the experimental limitations to characterize these structures, an obvious solution to the problem could be constituted by computational approaches [2,4,5,7,8,9,10]. Although some significant success in understanding receptor-antigen recognition has been achieved through the application of homology modeling approaches [11], the development of novel machine-learning-based predictive tools, such as those implemented in AlphaFold (AF), provides new opportunities that can have a remarkable impact in the field [12,13].

In this framework, we exploited the potentiality of these new predictive approaches to characterize at the atomic level the recognition of two important epitopes by their known receptors. In particular, we evaluated the binding of the immunodominant epitope corresponding to the region 49–57 of the E7 protein of the Human Papilloma Virus (HPV) to H2-D^b^, a class I MHC [14]. Notably, a mutation in this antigen (N53S) abolished the presentation of the murine H2-D^b^-restricted HPV16 E7 peptide in a head and neck squamous cell carcinoma mouse model [15]. Moreover, it has been shown that the same mutation eliminates the immunogenicity of E7 and is responsible for the evasion of the mutated TC-1 clones from the E7-specific immune responses induced by vaccination [16]. We have recently shown that the optimization of this epitope may improve the antigen presentation and the anti-tumor response [17,18]. In addition to the E7 epitope, a second well-characterized mouse tumor antigen, expressed by the murine B16 melanoma, residues 180–188 of tyrosinase-related protein 2, TRP-2 [19], has been investigated using this computational protocol. The TRP-2 peptide is implicated in the immunotherapy of human and murine melanoma, as it is recognized by both human and murine cytotoxic T-cells and is presented by the MHC class I haplotypes HLA-A*0201 and H2-K^b^, respectively, leading to the effective induction of antitumor immunity. Several research studies have addressed the recognition patterns of this peptide showing different specificities to MHC binding, highlighting all the factors that can impact the immunological responses to peptides [18,20,21,22]. In this framework, we report a structural and dynamic-based strategy to gain insights into the molecular basis that drives the recognition and the interaction of MHC class I in the immune response triggered by pathogen- and/or tumor-derived peptides.

Our findings demonstrate that not only can the approach here applied generate a reliable model for the complex formed by the E7 peptide and H2-D^b^, but it can also discriminate between different binding modes of the same epitope to different receptors. In this light, this protocol offers a valuable, low-demand, and fast research solution to obtain very accurate structural models of highly variable molecular complexes. Finally, based on the information collected here, we were able to rationalize the observed preferences of the studied receptors for specific residues in certain positions of the antigen sequences.

## 2. Results

We investigated the molecular basis of the structural recognition triggered by different MHC class I molecules by means of ab initio predictions and molecular dynamics (MD) simulations. **RAHYNIVTF** is a linear peptidic epitope (epitope ID 53112) studied as part of the protein E7 from HPV, hereafter referred to as **E7**. Recent findings have shown that the E7 peptide interacts with the MHC H2-D^b^ molecule with a high affinity [14,17]. Antigen processing and presentation are highly dynamic mechanisms that render the structural characterization of the complex a challenging task. Table 1 lists the structural models of the peptide epitopes bound to MHC molecules obtained by AF predictions (see Section 4).

### 2.1. Quality Assessment of AlphaFold Predictions

#### 2.1.1. GP33D^b^ System

To test the reliability of the AF predictions, we modeled the structural complex of the murine class I MHC molecule H2-D^b^ bound to the GP33 peptidic epitope KAVYNFATC, from Lymphocytic Choriomeningitis Mammarenavirus (LCMV), for which the X-ray structure was available (PDB entry: 1FG2 [23]). We calculated the structure of GP33^Db^ using the AF protocol that makes no use of the template’s information to minimize prediction biases, as illustrated in the Methods section. The evaluation of the AlphaFold error estimates evidences high-confidence structural predictions, showing on average a pLDDT value > 70, considering the complex model (peptide-MHC protein), see also the PAE in the Appendix A.

The large agreement between the experimental structure and the AF model is shown in Figure 1. The root mean square deviations (RMSD) value between the H2-D^b^ C^α^ atoms is 0.79 Å, reaching 1.40 Å when all atoms are considered (RMSD_aa_ = 1.25 Å for all atoms and RMSD_Cα_ = 0.78 Å when only secondary-structure elements are taken into account).

For the epitope, a perfect superposition is evident for the GP33 C-terminal portion (residues 7–9), at the F pocket, while minor differences can be observed at the N-terminal side. Different side chain rotamers are observable for the epitope residues K1, in the A pocket, and Y3 (Figure 1), with RMSD_Cα_ = 0.53 Å (RMSD_bb_ = 0.68 Å for backbone atoms, RMSD_aa_ = 1.51 Å).

The analysis of the interactions that mediate the recognition and the binding in the GP33^Db^ X-ray structure reveals several hydrogen bonds that concur to stabilize this interaction, involving the main or side chain atoms from either the epitope or the MHC protein (Table 2).

#### 2.1.2. TRP-2^A68^ System

**SVYDFFVWL** is a linear peptidic epitope (epitope ID 62404) studied as part of L-dopachrome tautomerase from *Homo sapiens* (human) and L-dopachrome tautomerase from *Mus musculus* (mouse), hereafter referred to as **TRP-2** (Tyrosinase-related protein-2).

TRP-2, specific for the mouse melanoma B16F10 cells, has been recently used as a model system to improve MHC-I affinity and TCR specificity [24]. A specific mutation at Position 4 demonstrated an enhanced binding to MHC-I molecules. The study pointed to the role of each of the amino acids of the epitope in the recognition and anchoring of both MHC-I and TCR proteins. To further investigate this aspect, we focus on the structural and physico-chemical determinants at the basis of this interaction. From the query of the IEDB database (https://www.iedb.org/ accessed on 1 November 2023), TRP-2 displayed a positive association with HLA (HLA-A*68:02, HLA-A*02:01), H2-D^b^, and H2-K^b^ molecules, when tested in several MHC binding assays [24,25,26,27]. Therefore, an analog modeling procedure was applied to validate the AF prediction on TRP-2 epitope binding to the murine class I MHC HLA-A68 (Table 1, TRP-2^A68^), before generating ex novo interaction models (TRP-2^Db^, TRP-2^Kb^). Here, the alignment between the X-ray structure (PDB entry: 4HX1 [24]) and the AF predicted model shows that the atomic positions of the two complexes deviate with RMSD_aa_ = 2.4 Å, RMSD_Cα_ = 1.65 Å for the HLA-A68 molecule, and RMSD_aa_ = 1.64 Å, RMSD_Cα_ = 0.62 Å, and RMSD_bb_ = 0.70 Å for the epitope atoms (Figure 2).

Despite a perfectly superposable fold, HLA-A68, H2-D^b^, and H2-K^b^ evidence minimal sequence differences (Table 3). AF structural models were obtained with high-level confidence (first rank scores are reported in Table 1, see also Appendix A) and generated distinct TRP-2 binding modes: while the TRP-2^A68^ AF model reproduces a crystallographic epitope pose, stabilized by a similar network of interactions, for the other two H2-D^b^ and H2-K^b^ molecules the TRP-2 epitope is stabilized into the MHC groove with a different pattern. Most of the variability appears at the TRP-2 middle portion, whereby an improved hydrophobic stabilization at the β-sheet level characterizes the H2-K^b^ complex. In all cases, the solvent-exposed S1 residue is anchored by H-bond interactions, mediated by the side chain atoms of amino acid residues pointing out of the cavity.

### 2.2. Ab Initio Prediction of E7 Interaction with H2D^b^

#### 2.2.1. AF Modeling: Evaluation of the Structural Models

The results obtained for the GP33^Db^ system prompt us to use it as a template to model the E7 binding to H2-D^b^. Analogously, the structural alignment between the crystallographic complex (PDB entry: 1FG2) and the AF-predicted model E7^Db^ (Figure 3) discloses the essential overlap of the two protein structures (H2-D^b^ RMSD_Cα_ = 1.35 Å, RMSD_aa_ = 1.73 Å), whereas the E7 epitope binds at the MHC groove displacing from the GP33 atoms with an RMSD_Cα_ of 0.47 Å (RMSD_bb_ = 0.48 Å). E7 recognition in the H2-D^b^ binding cavity is mediated by an equivalent pattern of interactions as observed for GP33, which is further stabilized by the packing contribution of H3 and F9 (Table 2). Minimal discrepancies evidenced in the stabilizing hydrogen bonds network between the experimental and predicted models may arise from the moderate resolution of the crystallographic structures and minor effects of the crystalline state on the one hand and expected minor errors associated with predictive algorithms on the other.

#### 2.2.2. A Dynamic View of the Interaction: MD Study

Molecular dynamics studies performed on the E7^Db^ AF model display the overall stability of the structural complex. In the RMSD plots (Appendix A), the evolution of the atomic deviations of MHC proteins from the starting structures rapidly stabilizes (<RMSD_Cα_ > = 2 Å) and rarely reaches RMSD_Cα_ values > 3 Å. Within the binding groove, E7 appears to be extremely stable along the simulations, the atomic deviations against the starting model of E7 reach RMSD_Cα_ average values of about 0.6 Å (Appendix A). The epitope atomic fluctuations analysis (root mean square fluctuation, RMSF) highlights the different mobility of the nonapeptide amino acid residues in the binding cavity. Indeed, although the C^α^ fluctuations are minimal, a fine structural modulation can be appreciated for the peptide side chains: the lowest fluctuation rate is associated with the fifth position (RMSF < 1 Å) enclosed by two adjacent more flexible amino acid residues, while the first position (R1) appears endowed with a higher variability than the last one (F9) (Appendix A).

In line with this result, the analysis of the backbone conformations (ϕ, ψ dihedral angles) adopted by the epitope residues throughout the simulation timescales confirms the high stability of the peptide spine in an extended shape within the MHC groove (the only exception is given by I6, found in the helical conformational space) (Appendix A). Overall, this stable anchoring is confirmed by the analysis of the distances between the centers of mass of the MHC and the E7 termini, R1 and F9 (Appendix A).

From the sequence alignment shown in Figure 3c, Position 5 is occupied by an asparagine residue in both epitopes, GP33 and E7. The two mobile and bulkier residues surrounding N5 in E7, namely Y4 and I6, do not affect the N5 anchoring at the MHC pocket. Indeed, the analysis of the interactions established by this residue along the MD simulations shows a persistent binding of its side chain with a pair of glutamine residues in both molecular complexes. N5 is held in place by stable and interchangeable hydrogen bonds established with the Oδ1 and Nε2 atoms of both Q71 and Q98 (Figure 4, Figure 5 and Appendix A).

During the simulation time, a significant number of interactions (hydrogen bonds, electrostatic and hydrophobic contacts), either transient or persistent, are made with the protein residues. Importantly, and in addition to the cardinal N5-mediated bonds, electrostatic interactions are established by the positively charged R1(E7^Db^) (Appendix A), and additional contacts, mediated by aromatic residues, stabilize the binding: namely, H3(E7^Db^) and T8 (E7^Db^) are bound to H156 and W148, respectively. In addition to the electrostatic stabilization mediated by the charged side chain of K147 with the terminal carboxyl group, the aromatic ring of the phenylalanine at the last position contributes to the improved packing of the C-terminal tail within the groove. F9 also establishes a hydrogen bond with OH-Y85 in the large aromatic cavity made by Y119, Y124, F117, W74, and W148 at the F pocket (Appendix A).

### 2.3. Ab Initio Prediction of TRP-2 Interaction with MHC Receptors

#### 2.3.1. Selection of MHC-Peptide Systems

The TRP-2 tumor epitope, SVYDFFVWL, is known to bind with low affinity to both mouse and human MHC class molecules. Some MHC ligand assays showed TRP-2 binding to H2-D^b^, H2-K^b^, and HLA-A* MHC receptors [24,25,26,27], and the crystal structure of TRP-2 in complex with the human HLA-A68 allele provides significant insights into the recognition mechanism of this peptide [24]. We ran AF structural predictions of the three complexes that TRP-2 forms with HLA-A68, H2-D^b^, and H2-K^b^ (Table 1).

#### 2.3.2. AF Modeling: Evaluation of the Structural Models

AF predictions of the TRP-2 epitope binding to human HLA-A68 and additional murine MHC class I molecules produced structural complexes endowed with a high confidence score, showing a good prediction quality, with an average pLLDT value > 70 and low predicted average errors, PAE, with some degree of variability localized at the epitope level (see Appendix A).

#### 2.3.3. A Dynamic View of the Interaction: MD Study

MD simulations highlighted an enhanced structural variability of the three TRP-2 complexes during the simulation time. First, a global structural inspection reveals that depending on the MHC molecule, TRP-2 amino acid residues present a diverse mobility within the binding groove (Appendix A), and a common tendency to displace from their starting position, especially at the N-terminus. Yet, we can appreciate a major displacement for TRP-2^Db^, which spans all along the nonapeptide, involving atoms from both the main and the side chains. Moreover, the extended conformation is compromised at the N-terminal portion of the epitope bound at the H2-D^b^ cavity, as displayed by the distribution of backbone dihedral angles. Defective anchoring is also observed for epitope S1 in TRP-2^Kb^, which is recovered in one of the two runs but still shows a lower fluctuation (Appendix A).

To better evaluate the quality of the epitope recognition, we followed the evolution of the hydrogen bonds observed in the AF-predicted models (Table 3). For all complexes, the first and the last epitope positions occupy specific locations at the A and F pockets (not necessarily stabilized by direct interactions). While observing a conserved pattern for the HLA model compared to the X-ray structure, we also found an improved stabilization for the H2-K^b^ at the central portion of the epitope. Throughout the simulations, weak interactions are lost, and other contacts are made to stabilize the binding. In all systems, the small polar S1 at the N-terminus does not establish a stable binding within the A-pocket, and the starting interactions (Table 3) are broken within the first frames of the simulation time. On the opposite side, L9 represents the major anchor at the F-pocket, with its carboxyl group electrostatically fixed by K147 and its hydrophobic side chain shielded in the nonpolar F pocket. The three complexes are stabilized by either hydrogen bonds or nonpolar contacts (Table 3). In Appendix A, we report the time evolution of the most stable interactions. Among all complexes, in TRP-2^Kb^ h-bonds and nonpolar contacts are well preserved along the simulation time. We must underscore that H2-D^b^ presents some differences at the α1 level, which may affect epitope binding (---**W**FRV**S**---): W74 and S78 are replaced by S71/T74 and D75/D78 in H2-K^b^ and HLA-A68, respectively. In H2-D^b^, W74 concurs to the hydrophobic F pocket that stabilizes the epitope C-terminal recognition, while in H2-K^b^ and HLA-A68 D75/D78 are directly involved in the stabilization of the L9 residue (Table 3 and Appendix A).

Again, for all MHC complexes, the main lock for the interaction is localized at the C-terminal side of the epitope. The analysis of the amino acid frequencies carried out for the three MHC molecules confirms these considerations. From Figure 6, we observe that the presence of a hydrophobic anchor at Position 9 (Val, Ile, Leu) is common to the three MHC alleles analyzed, while no favored amino acid emerges at Position 1. A large preference is shown by an Asn at Position 5 in the case of H2-D^b^, a Thr (and Val) preferably occupies the second position in the case of HLA-A68. On the other hand, H2-K^b^ exhibits a preference for binding for peptides endowed with an aromatic core (Tyr3 and Phe5/Phe6).

## 3. Discussion

The remarkable divergence of the MHC proteins and the chemical diversity of the exposed epitope make the generalization on receptor-antigen recognition extremely difficult. The necessity to specifically characterize each of these potential binary complexes would greatly benefit from the development of standardized predictive computation approaches.

By exploiting the huge potential of AlphaFold, we here investigated the hitherto uncharacterized binding mode of the widely used E7 epitope to its H2-D^b^ receptor combining this predictive approach with MD simulations. The high confidence of AF structural models of the peptide-MHC interaction has been further evaluated by running an ab initio prediction of the binding of the N53S mutant to the same H2-D^b^ receptor, which eliminates the immunogenicity of the E7 protein [16]. This AF test generated complexes with undermined peptide-receptor contacts, as demonstrated by the PAE matrices (Appendix A). Moreover, in the best-ranked model of the complexes made by the mutant, the peptide adopts a different conformation compared to the *wild-type* E7 peptide, with the serine pointing out of the cleft. In addition, AF predictions for the MHC complexes made by H2-D^b^ with either the reverse E7 sequence or a non-binder epitope [28] produced badly scored structural models. In Appendix A, the PAE matrices obtained for the strong-binder RAHYNIVTF and its reverse non-binder FTVINYHAR clearly evidence epitope-binding properties, emerging as key screening tools for further investigations.

Dynamics-based analyses clearly demonstrate the crucial role of the residue at Position 5 of the E7 peptide in recognition. Indeed, the side chain of the N5 residue forms a stable hydrogen bonding interaction with the side chains of Q71 and Q98 of the receptor. Stable hydrophobic interactions established by C-terminal residues of the peptide also contribute to the binding. Interestingly, the analysis of the observed frequencies of the residue at the fifth position for the nonapeptide epitopes specifically anchored by H2-D^b^ highlights a strong preference for the Asn residue in Position 5 (Figure 6). Moreover, in line with the present modeling, the hydrophobic residues are clearly overrepresented in the C-terminus of the epitope.

In the attempt to expand the implications of the present study, we searched the PDB looking for complexes formed by H2-D^b^. This survey led to the identification of 23 PDB epitope peptides complexed either with H2-D^b^ or its variants (sequence identity > 99%). The analysis of the sequences and the binding modes of these peptides into the receptor groove indicates that they share conserved patterns. In line with our findings, the comparative analysis of the sequences indicates the prevalence of the Asn residue in Position 5 and of apolar residues in Positions 3 and 9 (Appendix A). A collective analysis of the binding modes shows a recurrent alternate motif of amino acid residues pointing in and out of the groove delimited by the two α-helices. Residues that are mostly buried upon H2-D^b^ receptor binding are those located in Positions 2, 3, 5, and 9. A significant exception to this trend is represented by the binding of the peptide FAPGVFPYM which is one of the very few cases in which Position 5 is not occupied by an Asn residue. As shown in Figure 7b, in contrast to the general trend, the valine in Position 5 protrudes outward. This is favored by the impossibility of the side chain of this residue of making H-bonding interactions with Gln71 and Gln98 and, possibly, by some conformational restriction imposed by the presence of two proline residues in its sequence. In any case, the presence of Ala in Position 2 and Met in Position 9 in this peptide favors the anchoring to the receptor.

Notably, a quite different picture emerges from the analysis of the binding of the TRP-2 epitope to its known receptors. Indeed, we observe a rigid binding for all residues of the peptide only for the murine receptor H2-K^b^, whose interaction with TRP-2 has been reported in multiple studies [27,29]. On the other hand, for receptors (HLA-A68 and H2-D^b^) whose interaction with this peptide has been reported only occasionally, we observe that only the C-terminus of the peptides stably anchors the F pocket of the partners, the N-terminus being quite flexible (Figure 7a). Again, these observations are fully compatible with the amino acid position-dependent frequencies detected for these MHCs. The global rigid binding of TRP-2 to H2-K^b^ is likely due to the presence of residues that are favored at Position 3 (Tyr), 5–6 (Phe-Phe), and 9 (Leu). Of these important hot spots, only the hydrophobicity at the C-terminus is also important for HLA-A68 H2-D^b^. Along this line, the reduced affinity of TRP-2 compared to E7 for H2-D^b^ may be ascribed to the lack of an Asn residue in Position 5 in the former.

These observations strongly support the binding picture that emerged from this study. In the current broad scenario of in silico predictive methodologies, with general or specific applications [10], our computational strategy represents an additional and valuable solution to tackle intricate structural problems, such as the highly variable TCR-peptide/MHC complex recognition process. Collectively, our data indicate the reliability of the approach here applied, as well as for detecting differences in the binding in single amino acids replacements. This finding is not trivial, as AlphaFold is often unable to predict the impact of single-residue mutations [13], but it turned out to be very sensitive to capture important differences when direct and strong interactions guide the anchoring of the epitope to the receptor in these systems. Therefore, this approach can be straightforwardly extended to large-scale analyses of antigen-epitope recognition and also as a valuable tool to complement experimental data.

**Figure 7 ijms-25-01384-f007:**
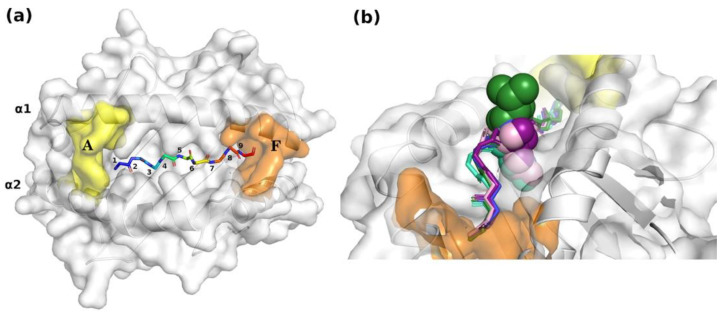
MHC-binding groove. (**a**) Surface representation of the A and F pockets and localization of the binding nonapeptide into the groove. For clarity, only epitope backbone atoms are reported in rainbow balls and sticks and numbered. White cartoons of the MHC protein (H2-D^b^) are visible in the background. (**b**) Close-up view of the structural alignment of representative peptides bound to the H2-D^b^ groove (Appendix A): KAVYNFATC (PDB ID: 1FG2, blue); FAPGVFPYM (PDB ID: 1BZ9 [30], green); LSLRNPILV (PDB ID: 3BUY [31], pink); ASNENIETM (PDB ID: 4HV8 [32], purple); SQLLNAKYL (PDB ID: 5WLG [33], cyan). Main chain atoms of the epitope amino acid residues are displayed in sticks, while all atom residues at Position 5 are rendered in spheres.

## 4. Materials and Methods

Selection of oncoprotein epitopes and generation of the MHC-presenting antigen complexes. The modified heteroclitic TAAs HPV-E7 and TRP-2 peptides (E7: RAHYNIVTF; TRP-2: SVYDFFVWL) displayed improved affinity to MHC-I molecules (H2-D^b^) and were immunologically validated in a mouse model. To date, no structural information on the two complexes (H2-D^b^/HPV-E7 and H2-D^b^/TRP-2) is available.

E7 peptide. Searches performed over the entire structural content of the PDB (https://www.rcsb.org/ accessed on 1 June 2023) using the sequence of the epitope (RAHYNIVTF) did not provide any results. In this scenario, the PDB was surveyed for complexes of H2-D^b^ with peptides sharing some sequence similarity with the HPV-16 E7 epitope^49–57^. A promising candidate was the PDB entry 1FG2 which corresponds to a complex of H2-D^b^ with the GP33 peptide (KAVYNFATC) [23].

TRP-2 peptide. Searches performed over the PDB using the sequence of the epitope (SVYDFFVWL) returned the X-ray structure of the tumor antigen-derived peptide bound to the human HLA-A68 (PDB entry: 4HX1 [24]). In this structure, the HLA-A68 molecule presents 21 amino acid mutations compared to the wild-type protein (UniProtKB ID P04439). For comparative purposes, we decided to consider the sequence of the mutated protein for the AlphaFold prediction of the complex. Additional structural predictions were also run for the UniProt reference sequence, confirming equivalent results.

AlphaFold predictions. The crystal structure of the peptidic epitope GP33 in complex with the murine class I MHC molecule H2-D^b^ (PDB entry: 1FG2) was used to train the AF protocol. Three-dimensional predictions of peptide-MHC complexes were obtained by the AF algorithm [13,34] by selecting the AlphaFold2-multimer-v2 as implemented on the Colab server (https://colab.research.google.com/github/sokrypton/ColabFold/blob/main/AlphaFold2.ipynb, accessed on 1 April 2022). Only the α1/α2-domain of the H-2 class I MHC (181 residues) was used for epitope binding, since this region is known as the recognition region of the peptide antigen. Specifically, the numbering of the α1 and α2 domains of H2-D^b^ (UniProtKB ID P01899) has been used as a reference, where the starting residue, Gly2, in our models corresponds to Gly25 of the UniProt entry.

Predictions were performed without considering any homologous experimental template (template_mode: none) using 24 recycles, and the resulting structures were relaxed with Amber Forcefield. The best predicted model (Rank 1) out of the five computed by AF for each complex is considered in the present study. The reliability of the AF models was assessed by evaluating the local distance difference test (LDDT) score. Details about the MHC complexes reported in this study are in Table 1 (Appendix A).


Survey of H2-D^b^-epitope structural complexes.


The inspection of the PDB in search of structural data of complexes formed by the class I MHC molecule H2-D^b^ with different epitopes was carried out in Protein BLAST (Basic Local Alignment Search Tool—https://blast.ncbi.nlm.nih.gov/Blast.cgi accessed on 1 November 2023) using the amino acid sequence of H2-D^b^ (UniProtKB ID P01899) as a query. This search led to the identification of 88 PDB entries sharing at least 80% sequence identity (coverage > 70%) with the input sequence. These structures were individually inspected to select only those containing complexes with epitope peptides. This procedure ended up with the identification of 74 entries of MHC-epitope complexes. The list of the PDB codes together with other details (epitope sequence and length, coverage, and resolution) are reported in the Appendix A.

Molecular Dynamics Simulations. MD simulations were run on the models reported in Table 1, using the GROMACS (v.2021.3) [35] software package with the Amber99sb force field [36], as in recent applications [37,38]. All structural models were immersed in triclinic boxes filled with water molecules (TIP3P water model [39]) and counterions (Na+ or Cl-) to balance charges. The simulations were carried out applying periodic boundary conditions. Systems were first energy-minimized for 50,000 steps using a steepest descent algorithm. Equilibration of each system was first conducted for 500 ps at a 300 K temperature (NVT ensemble) and then for 500 ps at 1 atm of pressure (NPT ensemble). After equilibration, production MD runs were performed for 200 ns on two independent replicas for each system (5 systems × 200 ns × 2 replicas), amounting to a total sampling time of 2 µs. The Parrinello-Rahman and the velocity rescaling methods [40,41] were used for pressure and temperature control, respectively. The particle mesh Ewald (PME) [42] with a grid spacing of 1.6 Å was used to compute the electrostatic interactions. For Lennard-Jones interactions, a cut-off of 10 Å was applied. Bond lengths were constrained using the LINCS algorithm [43]. An integration time step of 2 fs was used.

The H-bonding interactions in the starting MHC-epitope structures were computed using the LigPlot+ software (v.2.2.8) [44]. The align routine of PyMOL (zero cycles) has been used to compute RMSD values in the pairwise comparison of the models considered in this study [45].

Analysis of MD trajectories was performed by using GROMACS routines and the VMD program [46]. The gmx rms tool of the GROMACS software was used to compute the RMSD values of each structure of the MD trajectories compared to the reference starting model.

Epitope predictions. The Immune Epitope Database (IEDB—https://www.iedb.org/ accessed on 1 November 2023) was queried to collect available epitope information, including experimental data on antibody and T-cell epitopes. Nonapeptides analyses and predictions of human, mouse, and monkey MHC class I affinities for the epitopes were obtained by the NetMHC-4.0 server [47,48]. The visualization of position-dependent amino acid residues as logo motifs is also provided [49].

## Figures and Tables

**Figure 1 ijms-25-01384-f001:**
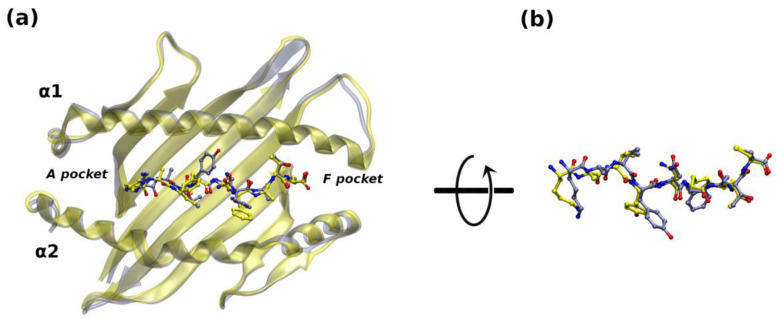
Structural alignment of the AlphaFold predicted models: (**a**) GP33^Db^ (gray) superposed on the X-ray GP33^Db^ structure (PDB entry: 1FG2) (yellow) is rendered with transparent cartoons; GP33 is displayed in ball and stick representation (N = blue; O = red); hydrogens are omitted. (**b**) Zoom in the epitope superposition after a 60° rotation around the *x*-axis.

**Figure 2 ijms-25-01384-f002:**
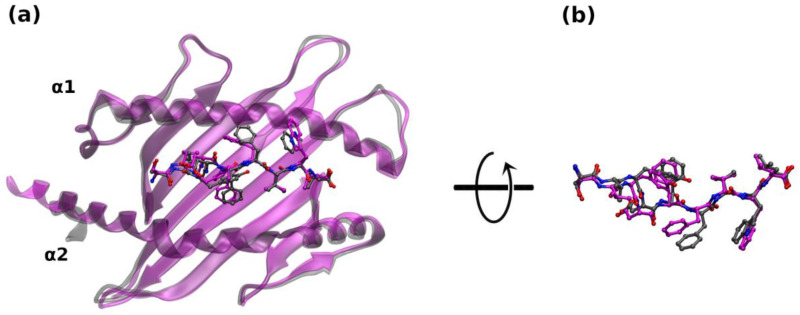
Structural alignment of the AlphaFold predicted models: (**a**) of TRP-2^A68^ in magenta with its corresponding X-ray structure (PDB entry: 4HX1) in gray cartoons; TRP-2 is displayed in ball and stick representation (N = blue; O = red); hydrogens are omitted. (**b**) Zoom in the epitope superposition after a 60° rotation around the *x*-axis.

**Figure 3 ijms-25-01384-f003:**
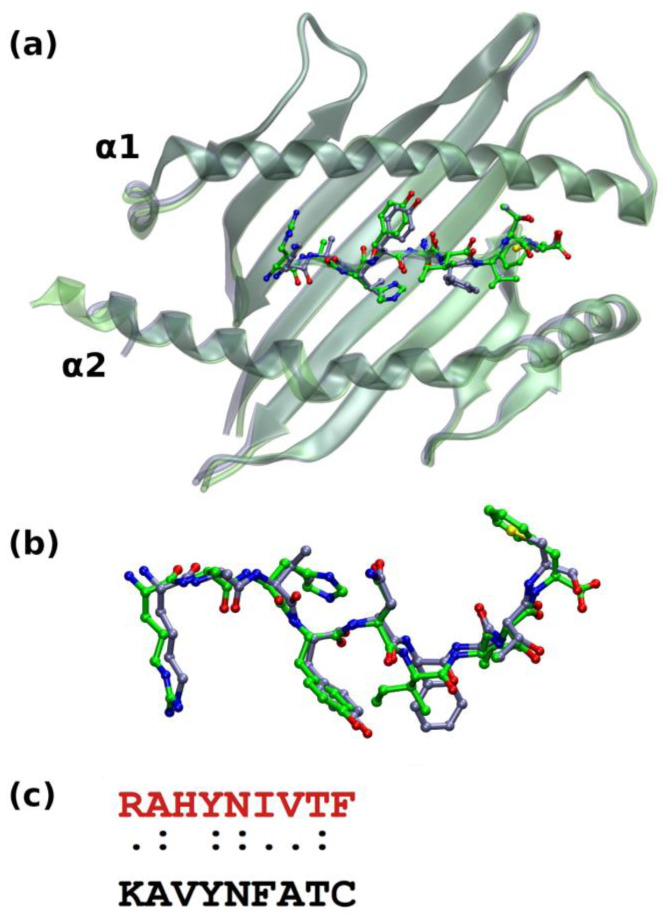
Structural alignment of the AlphaFold predicted models: (**a**) the systems GP33^Db^ and E7^Db^ are colored blue and green, respectively; H2-D^b^ is rendered in transparent cartoons, and nonapeptide GP33 and E7 are shown in balls and sticks using for the C atoms the same color codes of the corresponding receptor (N = blue; O = red; S = yellow); hydrogens are omitted. α1 and α2 helices delimiting A and F pockets are indicated. (**b**) Zoom in the epitopes superposition after a 60° rotation around the *x*-axis. (**c**) Sequence alignment between HPV-16 E7 (red) and LCMV GP33 (black) epitopes.

**Figure 4 ijms-25-01384-f004:**
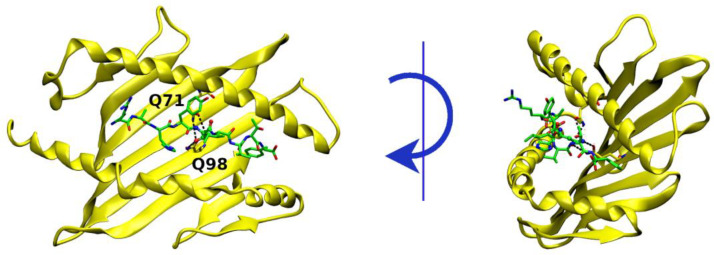
3D structure of E7^Db^. Two different orientations of the starting binding models are displayed (right: a *y*-axis 90° rotation is applied) and hydrogen bonds established by N5 with Q71 and Q98 are evidenced and labeled.

**Figure 5 ijms-25-01384-f005:**
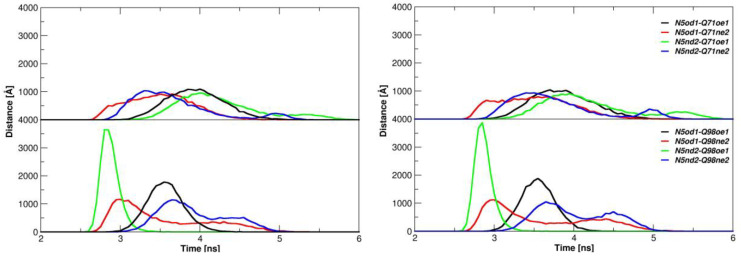
N5-Q71/Q98 hydrogen bonds. Distances among the epitope N5 and MHC Q71 and Q98 residues are reported for E7^Db^ for the two replicas (**left** and **right** panel).

**Figure 6 ijms-25-01384-f006:**
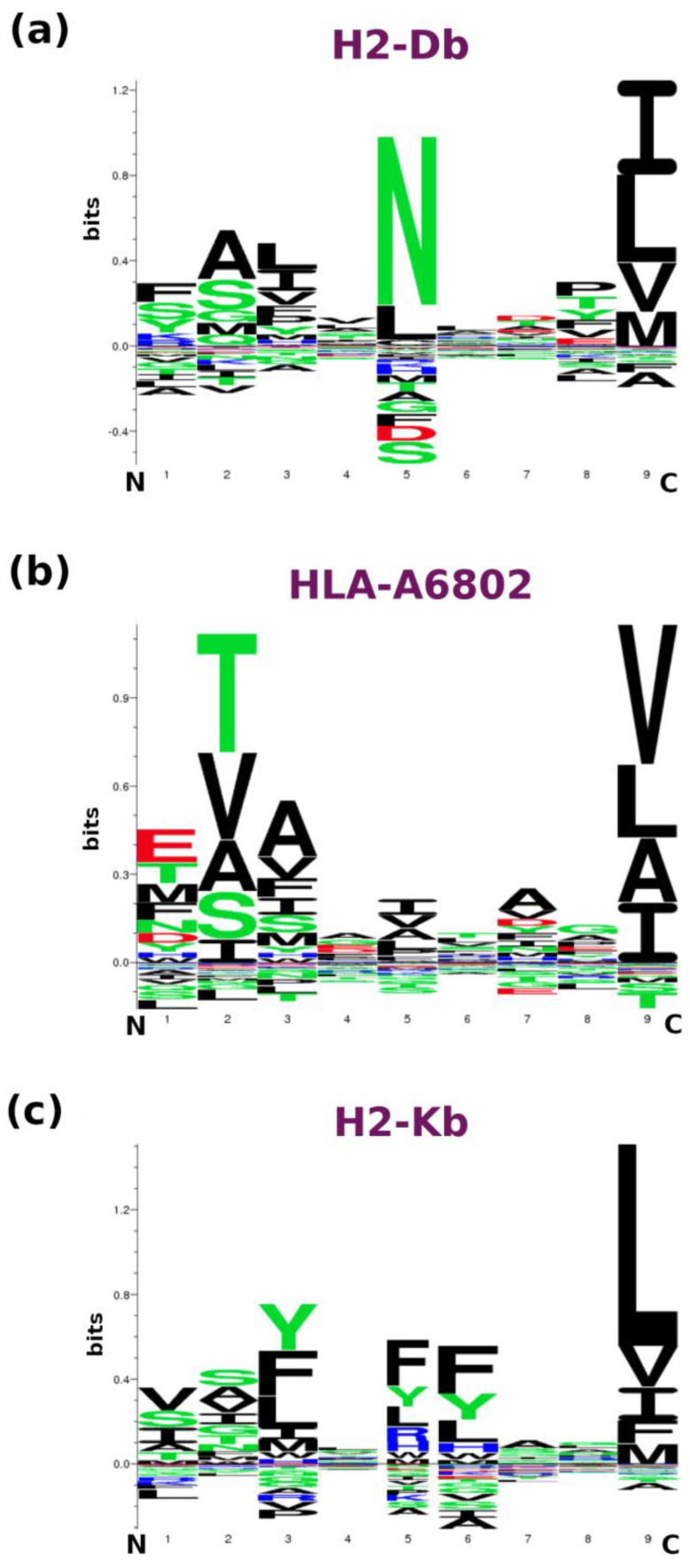
Propensities of binding of peptides to MHC class I molecules. LOGO motif representation is reported for nonapeptides binding to human murine alleles H2-D^b^ (**a**), human HLA-A68:02 (**b**), and H2-K^b^ (**c**).

**Table 1 ijms-25-01384-t001:** Structural models of peptide-MHC complexes reported in the present study.

EPITOPE SEQUENCE	H2 CLASS I MHC		
	**HLA-A68** UniProtKB ID P04439	**H2-D^b^** UniProtKB ID P01899	**H2-K^b^** UniProtKB ID P01901
**GP33** **KAVYNFATC**		GP33^Db^ * (1FG2)	
**E7** **RAHYNIVTF**		E7^Db^	
**TRP-2** **SVYDFFVWL**	TRP-2^A68^ *(4HX1)	TRP-2^Db^	TRP-2^Kb^

***** Models for which the crystal structure is available; the corresponding PDB entry is indicated in brackets. MD simulations were performed for underscored systems. For prediction quality assessments, see the PAE matrices in the Appendix A.

**Table 2 ijms-25-01384-t002:** Hydrogen bond interactions in the starting MHC-GP33/E7 structures.

GP33		H2-D^b^		E7		H2-D^b^
Residue	Atom	PDB ID 1FG2 *	AF Model	Residue	Atom	AF Model
**K1**	N	Y172OH	Y8OH	**R1**	N	Y8OH Y172OH
	O		Y160OH			
	NZ	E164OE2			NH	R63O
**A2**	N	E64OE1		**A2**		
**V3**	O		Q71NE2	**H3**	O	Q71NE2
					NE2	H156NE2
**Y4**	O	H156NE2	H156NE2	**Y4**		
**N5**	N	Q71OE1		**N5**	ND2	Q98OE1
	OD1	Q98OE1 Q98NE2	Y157OH Q98OE1 Q98NE2			
	ND2	Q98OE1 Q98NE2	Q71OE1 Q98OE1 Q98NE2			
**F6**	N	W74NE1		**I6**		
	O	Y157OH				
**A7**	O		W148NE1	**V7**	O	W148NE1
**T8**	O	W148NE1 K147NZ	W148NE1	**T8**	O	K147NZ
	OG1	K147NZ				
**C9**	N	S78OG	S78OG	**F9**	O	N81ND2 Y85OH K147NZ
	O		N81ND2			

* For comparative purposes, the sequence numbering of H2-D^b^ has been changed according to the UniProt numbering (see Section 4).

**Table 3 ijms-25-01384-t003:** Hydrogen bond interactions in the starting MHC-TRP-2 structures.

TRP-2		HLA-A68		H2-D^b^	H2-K^b^
Residue	Atom	PDB ID 4HX1	AF Model	AF Model	AF Model
**S1**	N	Y171OH Y7OH	Y171OH	E164OE	
	O	Y159OH	Y159OH		
	OG	R62NH N63ND2			K66NZ
**V2**	N	N63OD1	N63OD1		
	O		N66ND2		
**Y3**	N	Y99OH			
	O		N66ND2		
	OH	Q70OE1	R97NH		
**D4**	O				R155NH
**F5**	O				Q114NE2
**F6**	O				R155NH
**V7**	O				Y116OH
**W8**	O	W147NE1	W147NE1 K146NZ		W147NE1
**L9**	N	D77OD	D77OD		D77OD
	O	T143OG K146NZ	T143OG K146NZ	N81ND2 Y85OH	K146NZ

For Asp, Glu, and Arg residues, equivalent side chain atoms are indicated as OD/OE/NH, respectively.

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
