# Peer review of "Structural and Dynamic-Based Characterization of the Recognition Patterns of E7 and TRP-2 Epitopes by MHC Class I Receptors through Computational Approaches"

_ijms, 2024, doi:10.3390/ijms25031384_

Round 1
Reviewer 1 Report
Comments and Suggestions for Authors
1. Major corrections
The authors have employed AlphaFold and molecular dynamic (MD) to better understand the topography of peptide and MHC-I interactions. However, it is unclear how similarly scored NetMHC 4.0 peptides (RAHYNIVTF, 0.11 and KAVYNFATC, 0.088) can provide new insights in MHC-I peptide affinity and stability, which are related to immunogenicity (PMC4976001). It will be more important to use two dissimilar peptides (e.g., a binder versus a non-binder), and expand in the current manuscript on the appropriateness of Alphafold to exclude non-binding peptides, and MD simulation to exclude destabilising peptides.
Second, the use of AlphaFold to obtain high-resolution peptide interactions remains unconvincing when the RMSD Ca of the GP33Db and TRP-2 peptides are 0.8A and 3.9A, respectively when aligned to x-ray structures. This is consistent with pp 12 of 17, whereby the authors highlighted that AlphaFold is insensitive to point mutations.
Collectively, the authors have described several peptide to MHC-I interactions based on AlphaFold modelled structures. However, the description of H-bonding network at high resolution would need experimental validation e.g. H/D-exchange mass spectrometry to support the use of AlphaFold at its current level of sensitivity.
2. Minor correction
pp 10/17: The statement " we observe that the presence of a hydrophobic anchor at position 9 (Val, Ile, Leu) is common to all MHC alleles" is NOT true. Basic residues are preferred for MHC alleles such as A*11:01.
Reviewer 2 Report
Comments and Suggestions for Authors
The manuscript, titled, Structural and dynamic-based characterization of the 2 recognition patterns of E7 and TRP-2 epitopes by MHC class I 3 receptors through computational approaches, from Dr. Antonella Paladino’s group, presents in silico data that predict MHC class I-epitope complex structures using AlphaFold-based computation.
Overall, the manuscript is well-written, and the experimental approach and its outcomes are valuable to the field.
Minor points:
In either the Materials and Methods or Results section, include the source from which the MHC residues/atoms forming hydrogen bonds with the epitopes in the PDB columns of Tables 2 and 3 are derived. In the Discussion section, explain the potential reasons for discrepancies in hydrogen bond-forming atoms between PDB and AF models of H2-Db.
Describe in the Materials and Methods section how RMSD values are determined.
In the Discussion section, please cite the article (https://doi.org/10.3389/fimmu.2023.1285899) and reemphasize the non-template approach that your group took. Describe the potential advantages of the AlphaFold platform over the PANDORA and pDock docking methods.
Line 390. Typo, HLA-A68:02.
Round 2
Reviewer 1 Report
Comments and Suggestions for Authors
General comment
The authors have employed AF to investigate the static interactome between the peptide and MHC-I binding cleft. Using predicted peptide binders, the mode of MHC-I peptide binding is reasonably consistent. This includes the conserved interaction of asparagine (N5) in both epitopes GP33 (KAVYNFATC) and E7 (RAHYNIVTF) as the anchor residue in the MHC-I binding pocket. Additionally, the authors have further employed molecular dynamics, and demonstrated certain peptide flexibility upon MHC-I binding (Figure S4: Ramachandran plots).
Lastly, the authors have employed the PAE matrix (Fig S11) as a proxy for E7rev (FTVINYHAR, NetMHCpan 4.1 score 7.22), a predicted bad H2Db binder versus E7WT (RAHYNIVTF, NetMHCpan 4.1 score 0.011). Thus, PAE matrix could be used as a deterministic factor before analysing the AF model. The authors to consider including the PAE matrices (in supplementary) as main figure (s).
Minor correction:
Result section:
RAHYNIVFT is a linear peptide epitope --> should be RAHYNIVTF.
Author Response
We thank the Reviewer for his/her careful evaluation of our revised manuscript. We modified the text accordingly, also highlighting the important role of using the PAE matrices as valuable tools for further structural investigations in the Discussion section.